# Microbiomes of Blood-Feeding Arthropods: Genes Coding for Essential Nutrients and Relation to Vector Fitness and Pathogenic Infections. A Review

**DOI:** 10.3390/microorganisms9122433

**Published:** 2021-11-25

**Authors:** Daniel E. Sonenshine, Philip E. Stewart

**Affiliations:** 1Department of Biological Sciences, Old Dominion University, Norfolk, VA 23529, USA; 2Vector Molecular Biology Section, Laboratory of Malaria and Vector Research, National Institute of Allergy and Infectious Diseases, National Institutes of Health, Rockville, MD 20852, USA; 3Biology of Vector-Borne Viruses Section, Laboratory of Virology, National Institute of Allergy and Infectious Diseases, National Institutes of Health, Rocky Mountain Laboratories, Hamilton, MT 59840, USA; pestewart@niaid.nih.gov

**Keywords:** vector-borne diseases, eubacteria, symbiotic microbes, genomes, enteric microbes, endosymbionts

## Abstract

Background: Blood-feeding arthropods support a diverse array of symbiotic microbes, some of which facilitate host growth and development whereas others are detrimental to vector-borne pathogens. We found a common core constituency among the microbiota of 16 different arthropod blood-sucking disease vectors, including *Bacillaceae*, *Rickettsiaceae*, *Anaplasmataceae*, *Sphingomonadaceae*, *Enterobacteriaceae*, *Pseudomonadaceae*, *Moraxellaceae* and *Staphylococcaceae*. By comparing 21 genomes of common bacterial symbionts in blood-feeding vectors versus non-blooding insects, we found that certain enteric bacteria benefit their hosts by upregulating numerous genes coding for essential nutrients. Bacteria of blood-sucking vectors expressed significantly more genes (*p* < 0.001) coding for these essential nutrients than those of non-blooding insects. Moreover, compared to endosymbionts, the genomes of enteric bacteria also contained significantly more genes (*p* < 0.001) that code for the synthesis of essential amino acids and proteins that detoxify reactive oxygen species. In contrast, microbes in non-blood-feeding insects expressed few gene families coding for these nutrient categories. We also discuss specific midgut bacteria essential for the normal development of pathogens (e.g., *Leishmania*) versus others that were detrimental (e.g., bacterial toxins in mosquitoes lethal to *Plasmodium* spp.).

## 1. Introduction

Blood-feeding arthropods are the vectors of many of the most serious infectious diseases that have plagued humanity throughout history. Insect-borne diseases, such as malaria, plague, epidemic typhus, yellow fever and dengue fever, have killed millions and have often led to the collapse of armies or even entire civilizations [1]. Others, such as trypanosomiasis, Chagas disease and Leishmaniasis, have killed millions but also led to damaging morbidity among the disease survivors [2]. Tick-borne diseases, such as Lyme disease, which have reached epidemic proportions in recent years [3], are not as lethal as the preceding examples but can cause substantial morbidity. Even after the disease-causing bacteria have been cleared from the patients by antibiotics, some byproducts of microbial growth persist and can cause life changing morbidity, e.g., severe arthritis, for months or even years [4]. Vector-borne diseases of livestock, e.g., tick-borne babesiosis, theileriosis and other deadly tick-borne diseases, have severely compromised the nutritional needs of people in tropical and subtropical regions throughout the world [5].

The scientific and medical communities have directed most of their attention to the discovery and characterization of the many microbial pathogens responsible for these vector-borne infectious diseases. Not surprisingly, non-pathogenic microbiota were largely neglected. Recently, however, the role of these microbial symbionts in blood-sucking arthropods has received considerable attention, including diverse vector insects and ticks [6,7,8,9,10]. Knowledge of the complexity of the microbiomes of insect and acarine vectors has increased greatly in recent years thanks to the availability of powerful sequencing platforms, such as Next-Gen sequencing and/or similar technologies. The results of these investigations have made it possible to assess the importance of symbiotic bacterial phyla, families, and even distinct bacterial species for the health of blood-feeding insect and acarine vectors. We now realize that the association between bacteria and animals is a very ancient phenomenon, with most bacteria co-existing in a beneficial symbiotic relationship [11]. Moreover, in-depth analysis of changes in relative abundance of diverse microbiota during development, blood feeding and reproduction has enhanced our understanding of the contributions of these microbes to vector health (fitness) and even pathogen survival and transmission to vertebrate hosts. These vector insects and acarines acquire a portion of their microbiota from their vertebrate hosts [12] or from the external environment, but all share the same food source (blood) and often share similar host body conditions (e.g., temperature). Consequently, if blood were the predominant factor acting on these microbes, one might expect that there would be substantial similarity in the composition of their microbiomes. However, blood lacks some essential metabolites, e.g., thiamine, pyridoxine, folate and other B-vitamins [13], that are required for the parasite’s metabolism, and instead these ectoparasites must rely on their microbiota to compensate for these missing nutrients. Consequently, we may ask, do the same microbes provide these essential nutrients to most or all the different vector arthropods?

Although blood-sucking arthropods all share the same food source, i.e., blood, regardless of the different vertebrate groups upon which they feed, they do not all share the same off-host external environments or life cycles. Some vectors (e.g., mosquitoes, such as *Anopheles gambiae*, and sand flies, such as *Lutzyomia longpalpis*) occupy tropical biomes with only minor changes in the air or soil and water temperature, while others (e.g., the subtropical Lyme disease ticks, *Ixodes scapularis, I. pacificus, I. ricinus* and *I. persulcatus*) must contend with extreme environmental changes during their 2–3-year life cycle. Do the same or similar microbial families help their insect or acarine hosts adapt to these different environmental challenges? Do these same bacteria support or compromise the natural development and reproduction of their arthropod hosts? Do non-pathogenic microbes injected during blood feeding “prime” vertebrate host immune systems in ways that favor simultaneous pathogen transmission? Finally, how do the numerous non-pathogenic microorganisms affect the survival and transmission of the vector-borne pathogens [14]?

The microbiomes comprise protozoans and fungi, Archaea, viruses and bacteria. However, this review will focus exclusively on the bacteria. We will examine the roles of symbiotic bacteria by addressing the following questions:(1)Do all vector arthropods share the same families and/or genera of symbiotic bacteria or, instead, do they have their own bacterial communities that contribute to their biological needs?(2)Are certain symbiotic bacteria commonly found in many or most vector arthropods beneficial for vector health (i.e., mutualistic) while others are non-essential (commensal)?(3)Are the nutrients, detoxifying protection and immune factors obtained from symbionts of blood-feeding insects and ticks different from those in bacterial symbionts in plant feeding insects?(4)How does blood feeding affect the microbial community of the vector host?(5)Does the presence of other enterosymbiotic or endosymbiotic bacteria disrupt vector-borne pathogen colonization and transmission?

Comparisons were drawn from a survey of published literature on this subject. Comparisons also were made using 16S ribosomal gene sequences and protein sequences from selected family groups, e.g., Enterobacteriaceae and Pseudomonadaceae, for certain highly conserved genes (e.g., peptidoglycan). The results of our studies are supported by data tables, sequence alignments and phylogenetic trees, described in detail in the following sections of this article. 

## 2. Do All Blood-Feeding Insects and Acarines Share the Same Symbiotic Bacteria?

According to [8], “One or two primary symbiont species have been found to coevolve along with their host in each taxon and several secondary symbiont species are shared by all arthropod groups.” These authors also suggest that for the “gut microbiota, several bacterial symbionts genera are hosted in common.” Is this true?

To address this question, we have assembled a table of bacteria from 10 phyla reported in different blood-feeding arthropod vectors. Appendix A lists the genera of symbiotic bacteria infesting 18 different vector species, representing examples of sand flies (*Lutzomyia longipalpis, Phlebotomus papatasi*), different groups of mosquitoes (*A. gambiae, Aedes aegypti, Culex quinquefasciatus*), bugs (*Triatoma fuscipes*), tsetse flies (*Glossina fuscipes*), black flies (*Simulium damnosum*), hard ticks (*Dermacentor variabilis, I. scapularis, I. ricinus, Amblyomma americanum* and *Rhipicephalus microplus*), fleas (*Ctenocephalides felis*), lice (*Pediculus humanus*) and the fowl *mite (Dermanyssus gallinae*), all of which feed either on mammals and in some cases also on birds and reptiles. The honeybee parasite (*Varroa destructor*) is included as a non-blood-feeding arthropod contrast. We recognize that some genera identified in our review of the microbiome literature may be due to contaminated reagents used during the nucleic acid isolations or in the amplification process and may not naturally occur in many arthropod microbiota. Examples of these genera include *Delftia*, *Stenotrophomonas* and *Acinetobacter,* among others [15,16]. Since these microbes have not been definitively excluded from the published microbiomes identified in this review, we chose to retain them in our analysis.

When reviewing the microbiota of these blood-feeding arthropods, it is apparent that there is tremendous diversity in the composition of their microbial communities. Several vectors, e.g., the sand fly, *L. longipalpis*, exhibit a remarkably rich diversity of symbiont families and genera, with 110 different genera comprising 86 different families, 19 different classes and nine different phyla. However, all these diverse bacteria are not equally abundant. In adult *Lutzyomia* from the Mediterranean region, 57.7% of the midgut bacterial symbionts were from Proteobacteria (predominantly Enterobacteriaceae, but also including *Moraxellaceae*, *Pseudomonadaceae* and *Xanthomonadaceae*), while 23.9% were from Firmicutes (mainly Bacillacidae). A similar pattern was found in *Phlebotomus* sp. from this same region; 46.8% were from Proteobacteria (mostly Enterobacteriaceae) and 39.8% were from Firmicutes (99.9% *Bacillus*) [17]. Thus, species richness is a transient event, often related to life stage, diet and environmental factors. Only three phyla (Firmicutes, Actinobacteria and Proteobacteria) and eight bacterial families (*Bacillaceae*, *Rickettsiaceae*, *Anaplasmataceae*, *Enterobacteriaceae*, *Sphingomonadaceae*, *Moraxellaceae*, *Pseudomonadaceae* and *Staphylococcaceae*) are represented in almost all (>85%) of the arthropods examined, suggesting that these bacteria form a core community of mutualistic symbionts.

Among the mosquitoes, genus richness was greatest in *C. quinquefasciatus*, with 49 genera, comprising 23 families, 15 classes and six different phyla. In the kissing bug, *T. brasiliense*, genus richness was much more limited, with only 23 different bacterial genera, comprising 19 families, nine classes and five different phyla. Among the ticks, genus richness was greatest in *I. scapularis*, with 52 different genera, comprising 30 different families, 14 different classes and six different genera. The microbial communities of all other vector species examined exhibit a much greater paucity of genera and associated families or classes. Caution is needed in interpreting these results since it is possible that greater genus richness in the mosquito and the tick may be due to these species having been studied more intensely than most other arthropod vectors.
**Definitions of terms used in this manuscript:****Symbiosis**—as originally and broadly defined, refers to any interactions between organisms, including mutualism, commensalism, or parasitism.**Endosymbiont**—a cell engulfed by another and the stable association of the two (intracellular).**Enteric (ectosymbiont) bacteria**—extracellular bacteria that predominantly live within the midgut of the animal host.

Among ticks, genus richness varies substantially between the different tick species. However, a core group of bacteria, both enteric and endosymbionts, predominate. For definitions of the terms used in this manuscript, including symbiosis, endosymbiont and ectosymbiont (enteric bacteria), see the box provided. In the brown dog tick, *Rhipicephalus sanguineus* (S.L.), *Bacillus* especially predominates [18]. In *Ixodes scapularis* nymphs, *Pseudomonas* spp. are among the most abundant midgut residents [6,19]. Other abundant taxa included *Ochrobactrum* sp., *Agrobacterium* sp. and the family Xanthomonadaceae (order Xanthomonadales, Appendix A) were detected in rodent blood [20,21], suggesting they may be acquired during the blood meal [22].

Clearly, the answer to the question, do all these vector arthropods share the same microbial symbionts is both yes and no! Yes, there is a common core constituency of bacterial families (highlighted in yellow in Appendix A), but no, they do not all share the same bacterial genera, although there are many that are represented across diverse categories. Figure 1 is a Venn diagram that shows three clusters of microbial symbionts comparing the bacterial genera in a sand fly (*Lutzomyia longipalpis),* a mosquito (*Culex quinquefasciatus)* and a tick (*I. scapularis*). The mosquito, with 46 genera, shares 33 genera with the sand fly and 18 with the tick. The sand fly has the largest number of genera (107), many of which are different from each of the other arthropod vectors. In this comparison, the sand fly shares 24 genera with the tick, but only 11 genera with both the mosquito and the tick, namely, *Acinetobacter, Bacillus, Delftia, Enterobacter, Luteibacter, Methylobacterium, Pseudomonas, Rickettsia, Sphingomonas, Stenotrophomonas* and *Wolbachia*. The highest amount of symbiont sharing was between the mosquito and the sandfly (33). However, there also was considerable sharing between the tick and the mosquito (18) and the sandfly (24), as well as 11 symbionts shared between all three vectors—a remarkable finding given the long evolutionary separation between ticks and insects.

There are more than 90 bacterial families represented in Appendix A, yet almost all the vectors share only eight (highlighted in yellow) bacterial families that almost all vectors have in common, namely, the Rickettsiaceae, Anaplasmataceae, Enterobacteriaceae, Pseudomonadaceae, Moraxellaceae, Bacillaceae, Sphingomonadaceae and Staphylococcaceae. Among the genera in these eight families, the two that are virtually universally present are *Wolbachia,* present in all the different representative vector species (except *Triatoma braziliense*, although *Wolbachia* has been reported from other blood-feeding hemipterans [23]), and *Rickettsia*, present in all but *T. braziliense* and *Glossina fuscipes*. Also widespread are species of *Pseudomonas,* present in all but the flea, *Ctenocephalides felis* and the louse, *Pediculus humanus,* and *Acinetobacter,* present in all but *G. fuscipes* and *I. ricinus* and, finally, *Staphylococcus,* found in all vectors except *A. americanum*. Among the various hard tick species, the bacteria that all of these ticks have in common with most other haematophagus arthropods include *Rickettsia, Wolbachia, Pseudomonas, Sphingobacterium, Acinetobacter, Enterobacteriaceae, Bacilliaceae, Staphylococcus* and *Stenotrophomonas* [6]. How can these findings be explained? One answer concerns the difference between intracellular endosymbionts and enteric ectosymbionts. Another aspect refers to the vector life history, whether they are exclusive blood feeders, such as ticks and triatomine bugs, or whether their immature stages are free living (and exposed to a diverse microbiota that might infect their guts), such as mosquitoes and sand flies. Intracellular endosymbionts are more likely to be conserved and passed from generation to generation. In contrast, the compositions of the enteric bacterial populations are more likely to undergo substantial transitions during developmental changes and after blood feeding [14], e.g., in the mosquito, *Anopheles gambiae*, with a diverse microbiome in unfed adults, only four families of enteric Proteobacteria remained two days after a blood meal [24]. A similar phenomenon was found in the kissing bug, *Triatoma sordida*. Comparison of the bacteria in different nymphal stages and adults showed that bacteria of four different phyla (Actinobacteria, Bacteroidetes, Proteobacteria, Firmicutes) were present throughout the life cycle and were comprised of a core group of 12 genera, though there were clear differences among them. Genera of Firmicutes and Proteobacteria increased throughout development, while those of the other phyla diminished. Some genera were merely transient residents acquired from the local environment that were eliminated through defecation [25].

The distribution of microbiota in the body organs of vector arthropods has only recently begun to receive more intense scrutiny. Earlier studies of vector microbiota reported the diverse bacterial taxa from whole body extracts or, in many cases, bacteria inhabiting the midgut, since this was the first point of contact with the host’s internal body tissues. However, in recent years, it has become evident that some microbial symbionts are adapted to other internal organs of the vector arthropod host. One of the first studies of the microbiome of a specific internal organ revealed the presence of a spotted fever group, *Rickettsia* sp. (later identified as *R. buchneri*), in the ovary of *I. scapularis*, as well as other endosymbiotic bacteria in different ixodid and argasid ticks ovaries and malpighian tubules. Included were the Q-fever bacteria, *Coxiella burnetii, Franciscella* sp. and *Wolbachia persica* [26]. More recent investigations have confirmed the selective tissue tropisms of *Coxiella* sp. for the tick ovary and malpighian tubules in diverse tick species [14,27]. Other reports found *Francisella* sp. highly concentrated in the salivary glands of *Amblyomma maculatum* [28] and more extensive tissue tropisms for *R. buchneri* able to colonize both the ovary and salivary glands of *I. scapularis* [29].

The reasons for these mutualistic relationships are believed to be related to their contribution of essential nutrients and defense against oxidative stress. Moreover, we have come to appreciate that many of the microbes long known as human and animal pathogens are not injurious to the insect or tick vector hosts. Rather, they are either commensal or even mutualistic residents, tolerated by their vector hosts for the benefits they provide. This topic is discussed elsewhere in this review (Section 3 How do symbiotic bacteria contribute to the vector’s blood feeding habit?). Similar examples of highly selective tissue tropisms are also found in diverse insect vectors. Analysis of the reproductive organs of the two malaria mosquitoes, *Anopheles gambiae* and *A. coluzzii*, revealed a core microbiome comprising seven bacterial genera shared by all tissues, namely, *Staphylococcus*, *Corynebacterium*, *Geobacillus*, *Micrococcus*, *Acinetobacter*, *Pseudomonas* and unidentified Enterobacteriaceae genera. Other bacterial symbionts occasionally present in the reproductive organs of both male and female mosquito reproductive organs include *Thorsellia anophelis* and *Asaia* sp., *Spiroplasma* in both sexes, as well as *Wolbachia* sp. in both testes and ovaries and *Rickettsia* sp. only in male accessory glands. Note that some of the core bacteria are enteric (i.e., ectosymbionts) rather than endosymbiotic, specifically *Staphylococcus* and *Pseudomonas*. The authors suggest that several of the endosymbiotic bacteria, especially *Spiroplasma*, have the ability to manipulate mosquito reproductive success and also block *Plasmodium* development [30]. In kissing bugs, the salivary glands were found to support at least seven different bacterial genera, namely, *Burkholderia, Gordonia, Rhodococcus, Enterococcus, Proteus, Corynebacterium, Arsenophonus* and an unidentified species of Enterobacteriaceae [31]; the first four of those listed above are also enteric bacteria. Whether these non-pathogenic bacteria in the triatomine salivary glands also may induce antimicrobial activity in the mammalian hosts and possibly affect pathogen transmission by these bugs should be explored. 

A more restricted microbiome, limited to only three different genera, is found in tsetse flies, *Glossina morsitans*, comprising three endosymbionts, namely, *Wigglesworthia glossinidia,* resident in the anterior midgut and milk glands of these flies, *Sodalis glossinidius,* found in many different body tissues, and *Wolbachia* sp. infecting the reproductive organs. The nutritional benefits accorded to the tsetse fly host and the disruptive effects of *Wolbachia* on fly reproduction will be discussed elsewhere in this review [32].

In some arthropod vectors, endosymbionts invade specific body organs only during female reproductive activity. In the American dog tick, *D. variabilis*, *Rickettsia montanensis* invade the developing oocytes in response to rapidly increasing levels of the so-called molting hormone, 20-hydroxyecdysone (20-E) [33]. The adaptations used by these bacteria to invade the ovary and the significance of these processes for pathogen transmission to the tick’s progeny are discussed below (Section 6 Does the vector microbiome affect the survival and/or development of pathogenic microbes?). Another important question that needs to be considered is whether the bacterial symbionts that infect these diverse insect and tick vectors have remained very similar over the eons of evolutionary time or, instead, have diverged greatly during the millions of years that their vector hosts also have diverged. The origin of the Ixodida is hypothesized to have occurred in either the Paleozoic (~320 mya) [34] or, more broadly, likely during the Carboniferous period between 360 to 300 mya [35] or the Mesozoic (~180–220 mya) [35] era, long after the divergence of the Pancrustacea from the Chelicerata, around 530 mya [36], and from the insects, approximately 480 mya [37]. To address this question, protein alignments and phylogenetic trees were used to compare peptidoglycan synthetase (an essential protein in the bacterial cell wall) of bacteria within the genus *Pseudomonas.* Appendix A (maximum likelihood analysis) shows the consensus protein alignment among 19 *Pseudomonas* strains and species. Figure 2 shows the phylogenetic tree relationships for the peptidoglycan synthetase proteins among the 19 pseudomonads, including several arranged in accordance with their arthropod vector hosts. There was a high level of similarity for this protein: the consensus identity was 82.3%; the pairwise positive (BLSM62) was 87.6%. The outliers are *P. pleccoglossicida, P. yamanorum* and *P. aeruginosa* (KJJ17746). Overall, there is substantial evolutionary distance between the *Pseudomonas* species, with distances ranging from 38 in *P. alcaligenes* (no known vector) near the bottom of the tree, to 98, 99 and 100 among the species near the top (*P. pleccoglossidida* was excluded from the analysis because its distance was so extreme). However, despite the ancient divergence between the Insecta and the Ixodida [38,39] there was no clear separation of the bacteria by insect or acarine taxa.

A similar comparison was made for the peptidoglycan synthetase proteins of bacteria in the family Enterobacteriaceae. Figure 3 shows the phylogenetic relationships among 22 members of the Enterobacteriaceae, including members of the microbiota of the sand flies *L. longipalpis*, *Phlebotomus papatasi*, the tick *I. scapularis* and the mosquito *Anopheles gambiae*, using the maximum likelihood method. Statistical analysis (Appendix A, MEGA, amino acid alignment) showed 94% similarity among the 22 species in this bacterial family, although analysis of the tree using the neighbor joining method Geneious was only 46% (not shown). Comparing the phylogenetic similarities, we observed one group, comprising the genera *Proteus, Xenorhabdus, Serratia, Citrobacter* and *Trabulsiella* found in the two dipteran insects, *Lutzomyia longipalpis* and *Phlebotomus papatasi;* unexplained is *Escherichia coli*, found in the tick *I. scapularis,* which also aligned with the others in this group. A second grouping includes the genera *Kluyvera, Erwinia* and *Pantoea*, all found exclusively in the same dipteran insects, *L. longipalpis* and *P. papatasi.* A third group includes the genera *Salmonella*, *Klebsiella* and *Raoultella*, found in the three dipteran insects, *A. gambiae*, *L. longipalpis* and *P. papatasi.* A fourth group includes the genera *Enterobacter*, *Klebsiella*, *E.*
*coli* and *Shigella*, all found in both *P. papatasi* and *I. scapularis*. 

So, what can we infer from this phylogenetic analysis? Briefly, the data suggests that the gene coding for peptidoglycan synthetase is very highly conserved among the bacteria in this family, similar to that found in *Pseudomonas* sp. Most of the genera in this bacterial family are found in the three dipteran insects considered in this study. However, unexpected is the close evolutionary relationship of the bacteria, *Enterobacter*, *Klebsiella* and *E. coli* in the tick *I. scapularis* and the insect *P. papatasi*. In summary, the data suggests the gene is highly conserved, despite the ancient evolutionary divergence among these very different arthropod vectors; they all share species of Enterobacteriaceae, perhaps because members of this bacterial family are beneficial to vector health. Possible reasons for these benefits are discussed below. An alternative hypothesis, however, that we cannot exclude, is that these proteins are conserved because of some other selective pressure that we are unaware of.

## 3. How Do Symbiotic Bacteria Contribute to the Vector’s Blood Feeding Habits?

Blood-feeding arthropods must cope with a variety of vertebrate host defenses to gain access to this nutrient-rich food source. Once they have identified the target host using their remarkably efficient chemosensory systems [40,41,42,43], these ectoparasites must pierce the skin, differentiate blood from the tissues [44,45] and commence feeding. In addition, the parasite must overcome the array of host hemostatic defenses to imbibe blood, prevent coagulation and avoid or suppress host inflammatory reactions that would disable or kill it [46,47,48]. However, successful blood feeding presents additional problems. The blood meal is rich in proteins, free amino acids and reactive oxygen species (ROS), presenting major challenges, requiring rapid conversion of the amino acids into new proteins for nutritional use and detoxification of ROS to cope with redox stress [49]. DUOX-ROS enzymes (i.e., enzymes with both NADPH oxidase and peroxidase domains), when expressed, can lead to substantial increases in gut microbiota populations [50].

Blood feeding also activates the tick’s or insect’s immune system, regulated primarily by the IMD (immune deficiency) pathway, leading to expression of antimicrobial peptides (AMPs) and survival of those bacterial taxa resistant to these immune peptides [50]. Those enteric bacteria that have adapted to this changed microenvironment undergo an enormous increase in abundance, while others are eliminated or their occurrence is greatly reduced, leading to a major reduction in microbial diversity. Surviving bacterial species have biosynthetic pathways that enable them to cope with the oxidative stresses, iron toxicity and amino acid overload that is a consequence of blood feeding. They also contribute essential nutrients missing in vertebrate blood. Most important is that vertebrate blood is deficient in certain essential nutrients, especially B vitamins [51].

We searched the COG database and found genes coding for more than 69 different vitamins, cofactors and other related proteins that could be used to examine their occurrence in symbiotic bacteria infecting different hematophagous vectors (for a list of these vitamins and cofactors, see Appendix A). The blood of most vertebrate species almost always lacks the essential vitamin B-synthesizing enzymes and cofactors, e.g., biotin, folate and other derivatives. In the tick *I. pacificus*, a rickettsial endosymbiont, *Rickettsia* sp. G021, was found to have five different genes of the folate biosynthesis pathway, corresponding to the same genes found in *R. buchneri*, an endosymbiont of *I. scapularis* [52]. *R. buchneri* is also abundant in *I. scapularis*, likely due to its ability to furnish folate (B9), which the tick cannot obtain from its vertebrate hosts [53]. A similar phenomenon was found in mosquitoes (*A. aegypti*) in which certain bacteria, especially *E. coli*, contribute to folate biosynthesis and concomitant energy storage, enhancing larval development [54].

Figure 4 presents a Phylogenetic analysis of biotin synthetase among 20 different bacterial species, indicating that this gene is highly conserved among these diverse bacteria.

## 4. How Do the Microbial Symbionts Affect Vector Health and Reproduction and Contribute to Fitness of the Vector Host?

There is increasing evidence that certain microbial symbionts are essential for vector fitness and even survival. As noted previously, vertebrate blood is deficient in many important micronutrients, especially vitamins and enzyme cofactors, such as flavin adenine dinucleotide (FAD), Co-enzyme A (CoA), nicotinamide adenine dinucleotide phosphate (NADP+) and others. Mutualistic endosymbionts are believed to contribute to these missing micronutrients and, perhaps, in other cases, essential amino acids. Tsetse flies, e.g., *Glossina morsitans*, harbor two mutualistic bacteria, *Sodalis glossinidius* and *Wiggleworthia glossinidia* (Enterobacteriaceae). The latter bacterium is found in endosomes within specialized epithelial cells (bacteriocytes). Elimination of *W. glossinidia* results in retarded growth and reduced fecundity, suggesting an essential role in the biology of these insects. The genome of *W. glossinidia* has more than 60 genes coding for the biosynthesis of diverse co-factors of B vitamins, vitamins absent in the blood of the insect’s vertebrate hosts [55]. Elimination of *W. glossinidia* leads to loss of vitamin B6, a co-factor for enzymes needed for the metabolism of amino acids. Their absence also impairs the production of the co-factor S-adenosyl methionine (SAM), leading to loss of methylation reactions needed for amino acid synthesis and other metabolic pathways [56].

Another example of an important endosymbiont is *Coxiella*. Eradication of *Coxiella* from the tissues of the lone star tick *A. americanum* showed that these endosymbionts were critical for their survival and overall fitness [57]. In this tick, *Coxiella* sp. are ubiquitous throughout the tick’s body tissues and are believed essential for the tick’s survival [58]. Moreover, the presence of these microbial endosymbionts in the *A. americanum* salivary glands impairs transmission of *Ehrlichia chaffiensis* [58]. Similarly, in another example illustrating the mutualistic role of microbial endosymbionts, antibiotic (tetracycline) treatment of cattle ticks, *Rhipicephalus microplus*, greatly reduced the populations of *Coxiella*, which completely disrupted the maturation of tick nymphs, although it had little or no effect on reproductive activity or embryo development. However, in a separate study, *Coxellia* endosymbionts were found to be essential for the survival of *A*. *americanum*; antibiotic treatment of these ticks led to reduced reproductive fitness [57]. *Coxiella* sp. is the predominate component of the microbiota in this tick, comprising 98% of all 16S rRNA sequences in eggs, larvae and other life stages. In a different tick, *R. microplus*, detailed examination of the genome of the *Coxiella* endosymbiont revealed genes coding for proteins involved in biosynthetic pathways for the co-factors for flavin adenine dinucleotide (FAD) and coenzyme A (CoA), as well as vitamins B3, B5, B6, B7 and B9 [59]. However, in my own experience (DES), antibiotic eradication (doxycycline) of rickettsial endosymbionts (*Rickettsia montanensis*) of the American dog tick, *Dermacentor variabilis*, confirmed by PCR, had no effect on the long-term survival of a laboratory colony of this species (Sonenshine, unpublished). Similarly, a study of *I. pacificus* also showed that antibiotic elimination of the Rickettsia endosymbiont did not affect viability or fecundity [60]. In the Lyme disease tick, *I. scapularis*, antibiotic elimination of the endosymbiont *R. buchneri* had no apparent effect on larval hatching [61].

In contrast to the beneficial effects of various endosymbionts described above, some symbionts are harmful to their vector hosts. A representative example is seen in the chigger mites, *Leptotrombidium* sp., the vectors of the often-deadly scrub typhus disease (also known as tsutsugamushi disease), where the presence of the endosymbionts *Cardinium, Rickettsia* and *Wolbachia* led to the killing of males, parthenogenesis and cytological damage. These injurious effects led to sexual imbalances, reduced fecundity and overall loss of fitness. This also affected transmission of the rickettsial pathogen, *Orienta tsutsugamushi*, in some chigger species [62].

One possible contribution is the provision of Ankyrin, an essential regulatory protein, from *Wolbachia* sp. (YP_001975093), which we also found in the synganglion transcriptome of the soft tick, *Ornithodoros turicata*, as an Ankyrin repeat domain protein [63]. Another from the *Wolbachia* endosymbiont from the same tick transcriptome is a “pao retrotransposon peptidase” |(ZP_00374549) believed to regulate the transcriptionally active retrotransposon that encodes the polyprotein Gag-Pol. In olive fruit flies (*Bactrocera oleae*), higher transcriptional activity in male rather than female germline tissue may contribute to a better understanding of how *Wolbachia* endosymbionts affect sex ratios [64]. A similar role has been ascribed to *Arsenophonus* spp., which mostly affects the male germline, causing the killing of male embryos [65], and results in the distortion of sex-ratios in the affected tick populations.

In African mosquitoes (*Anopheles coluzzii*) that become infected with trypanosomes, *Trypanosoma brucei*, qPCR revealed a five-fold increase in the total enteric bacterial population within two days after infection and again after a blood meal. The effects were similar regardless of the blood source [66]. The intensity of *Plasmodium* infection in its mosquito host also may be directly correlated with the abundance of midgut Enterobacteriaceae populations. These bacteria significantly reduced the intensity and infection-prevalence of *Plasmodium falciparum* acquired via multiple infectious feedings. The authors speculate that oral infection with the bacteria may disrupt midgut homeostasis and subsequently activate immune responses to the parasites.

In summary, the significance of these findings is that the composition of the vector’s gut microbiota is one of the major components that may determine the success and probably the intensity of malarial infections in the mosquito (*A. gambiae*) vectors [67,68].

A total of 68 vitamins and vitamin-related proteins, (e.g., co-factors, transport proteins, transaminases, etc., Appendix A, based on COG database), as well as ROS regulating enzymes and antimicrobial peptides produced by microbial symbionts of blood-feeding and plant-feeding insects and ticks were compared to assess their relative contributions to host health. In addition, we created a list of 18 symbiont genomes (Appendix A) (subsequently expanded to 21 genomes, data not shown) including representative blood-feeding versus plant-feeding symbiotic bacteria, using PATRIC (www.Patricbrc.org, accessed on 24 June 2019) and queried these genomes to determine the presence and relative abundance of genes for these different nutrients. When we searched Protein Families for biotin synthase, the system showed 106 gene categories, including biotin synthase, dethiobiotin, etc. The enteric bacteria in blood feeding vectors were especially rich. *Rhodococcus rhodnii* (the endosymbiont of *Rhodnius prolixus*) had 11 different protein families, the most of any species in the group; others, e.g., *Pantoea agglomerans* had eight, *Stenotrophomonas maltophila* had seven, etc., but the genome for *Buchnera aphidicola*, a symbiont of aphids, had only two. A statistical comparison of the distribution of nutrient categories, ranging from biotin to folate, showed that the enterobacteria, i.e., bacteria in the lumen of the insect or tick midgut (or cavities of other body organs) had statistically significantly (*t* = 4.5–14) more such genes coding for these nutrients than non-blood feeding insects, ranging from 45–82 in the former, versus only 3–15 of these genes in the latter (Figure 5A). When analyzed by two-way ANOVA (Prism, www.Graphpad.com, accessed on 9 May 2021), the results showed that there was no significant difference among the genomes for the enteric bacteria (F 8, 80) 1.88, *p* > 0.08, but there was a highly significant difference between rows (F 10, 80) 9.66, *p* < 0.001). When the group analysis was done for all 21 genomes, blood-feeding versus non-blood-feeding insects and ticks, the differences between the columns was highly significant, (F 20, 220) 8.77, *p* < 0.001 as well as for all rows, F (11, 220) 11.45, *p* < 0.0001. When the analysis was done between the enteric versus the endosymbionts the results were highly significantly different, *p* < 0.0001 (F 17, 187). Similarly, as shown in Figure 5B (a comparison of genes coding for pimelate through xanthine), the conclusion seems inescapable that the bacterial symbionts in blood-feeding insects and ticks contribute significantly to the nutrient needs of these vectors, in contrast to the symbionts of non-blood feeding insects where there is a paucity of such genes. Moreover, following further investigation using two-way ANOVA, comparison of the nutrient contributions (vitamins) of each of the different bacteria in blood-feeding vectors versus the plant feeder *Blochmania chromaiodes* showed that seven out of the 10 enteric bacteria were significantly different. The exceptions, *Stenotrophomonas*, *Borrelia burgdorferi* and *Y. pestis*, were likely due to the low abundance of vitamin genes for these three bacteria (Appendix A: Dunnet’s test comparing gene frequencies in diverse enteric bacteria against the endosymbionts). In contrast, none of the bacterial endosymbionts of blood-feeding vectors were significantly different from the endosymbionts of plant-feeding insects, including *Blochmania*, *B. aphidicola* and *Cardinium* (Appendix A: Dunnet’s test comparing bacteria from blood-feeding vectors versus a non-blood feeder, *B. aphidicola*). Similarly, although gene frequencies in 11 of the enteric bacteria were highly significantly different from the endosymbiont *Rickettsia belli*, gene frequencies among the endosymbionts were not significantly different from *R. belli* (Appendix A: a comparison of enteric bacteria with an endosymbiont, *R. belli*, with two-way ANOVA results). However, as noted previously (Section 2 Do All Blood-Feeding Insects and Acarines Share the Same Symbiotic Bacteria?), the nutrients provided by such endosymbionts as *Wigglesworthia glossinidia* are highly specific, e.g., vitamin B6 and the co-factor S-adenosyl methionine (SAM).

A similar study was carried out comparing the genes encoding proteins that cope with cell stresses, such as reactive oxygen species (ROS), heat shock and reactive nitrogen species. Coding compacities of the same groups of bacteria in blood-feeding insects and ticks versus bacteria infecting non-blood-feeding insects were compared. Figure 6A,B shows the results in graphic form. ANOVA (one-way) results were highly significant (F = 6.284, *p* < 0.001, 20, 210 DF). When analyzed by two-way ANOVA, multiple comparisons, the results were highly significantly different for column comparisons, F 10.13, (20, 200 DF), *p* < 0.0001. When comparisons were drawn using Dunnet’s test for multiple comparisons of the genes encoding cell stress-related proteins in the different bacteria against the plant feeder *B. aphidicola*, 10 of the 14 enteric bacteria (*E. cloacae, Acetobacter, Acinetobacter, Staphylococcus aureus, Sphingomonas witchii, Serratia marcesans, Pantoea, Pseudomonas, Stentotrophoma* and *Rhodococcus*) had significantly more of these genes (Appendix A, yellow highlight) than the plant-feeding symbiont. Similar results were obtained with the Dunnet’s test with the genes encoding the cell stress proteins in different bacteria against the endosymbiont *R. bellii*. Here too, the genes from 11 of the same group of enteric bacteria noted above were significantly different than the endosymbiont. These findings support the hypothesis that most of the enteric bacteria contribute a significantly greater variety of genes encoding detoxifying proteins to blood-feeding vectors versus bacterial endosymbionts and symbiotic bacteria among the non-blood-feeding insects. Genes coding for oxidases and other detoxifying enzymes are clearly beneficial to their blood-feeding vector hosts. An example of this phenomenon is found in the results of a study that identified high-affinity cytochrome bd oxidase secreted by enteric *E. coli* which strongly lowered oxygen levels. The evidence supported the hypothesis that hypoxia enabled mosquito growth and ecdysone-induced molting [69].

We also compared the contributions of the essential amino acids that the enteric and endosymbiont bacteria of blood-feeding insects and ticks contribute versus those from the bacterial symbionts of non-blood-feeding insects. Figure 7A shows the results in graphic form. When analyzed using one-way ANOVA, the results were highly significant (F = 4.866, *p* < 0.0001, 17, 162 DF). When comparisons were made using Dunnet’s test for multiple comparison of the essential amino acids in the different bacteria against the plant feeder *B. aphidicola*, four of the 10 enteric bacteria, *E. cloacae*, *S. marcesans*, *Pantoea* and *Pseudomonas*, were highly significantly different from the plant feeder symbiont, *B. aphidicola* (Appendix A). However, when compared using the *t*-test, all other enteric bacteria were significantly different from *B. aphidicola*, namely, *Sphingomonas* (t = 2.28, *p* < 0.05), *Stenotrophomonas* (t = 2.82, *p* < 0.02) and *Rhodococcus* (t = 5.03, *p* < 0.001), as well as the other enteric bacteria previously shown to be significantly different. These findings provide additional support for the enteric bacteria providing essential nutritional benefits for the blood-feeding insect and tick vectors. Figure 7B shows the same data but with the ten different amino acids on the horizontal axis. Four amino acids, methionine, threonine, histidine and arginine, were the amino acids with the highest gene frequency.

Nevertheless, there are exceptions. Not all enteric bacteria infecting the midgut of insects or ticks appear essential for vector health. For example, in the sheep ked, *Melophagus ovinus*, several of the same enteric bacteria noted previously are also present in this insect, specifically, *Enterobacter, Acinetobacter* and *Staphylococcus*, but only in low abundance [70].

Non-blood feeding herbivorous insects also depend upon populations of obligate symbionts for essential nutrients missing in the plant hosts. One of the best examples of this relationship is the pea aphid, *Acyrthosiphon pisum*, and its dependence upon the γ-proteobacterium *Buchnera aphidicola* (Enterobacteriacae). These bacteria colonize specialized cells, known as bacteriocytes, in the midgut epithelial lining. *Buchnera* endosymbionts are involved in the production of essential amino acids by its aphid host during embryonic development, e.g., AroK, a protein involved in the biosynthesis of metabolites via the metabolite choirmate which in turn serve as precursors for the aromatic amino acids phenylalanine, tryptophan and tyrosine. *Buchnera* sp. also have genes that encode for proteins that are produced or co-produced by these bacteria at different stages of the aphid life cycle, including ornithine, histidine, lysine, threonine, leucine, tryptophan, phenylalanine, sulfur reduction and cysteine [71,72]. *Buchnera* also encodes genes involved in riboflavin production, while *Baumannia*, an endosymbiont of the sharpshooter (*Homalodisca vitripennis*), retains genes that enable production of several of the B-vitamins, and *Sulcia muelleri* generates most or all of the essential amino acids for the host [73].

## 5. How Does Blood Feeding Affect the Microbial Community of the Vector Host?

Blood feeding induces major changes in the bacterial composition of the vector’s microbiota. During blood feeding, survival of bacteria within the vector insect or tick is strongly dependent upon protection from oxidative stress. In ticks, e.g., the Gulf Coast Tick, *A. maculatum*, thioredoxin reductase (TrxR), a selenoprotein, is a major component of the antioxidant pathway that is essential for maintaining redox homeostasis balance within the tick host. A TrxR homologue was also found to be transcribed in the salivary glands of the mosquito, *A. gambiae*, suggesting that this system may be widespread among blood-feeding arthropods [74]. In ticks, e.g., *A. maculatum*, many of the bacterial species are eliminated during and after blood feeding. However, *Enterobacter* sp., *E. cloacae* and other Enterobacteriaceae predominate in tick midguts, after surviving the upregulation of genes encoding for ROS-detoxifying proteins and ROS-mediated killing action stimulated during the digestion of red blood cells. Though it is admittedly speculation, we hypothesize that these species survive only as long as ingested blood is still present and die out after the blood meal is depleted. They probably did not evolve their numerous ROS detoxifying enzymes as an adaptation to survival in the midgut; rather, they evolved for survival in the external environment and these enzymes allow them to take advantage and persist in the midgut environment temporarily with the availability of the blood meal nutrients (see Section 4, How Do the Microbial Symbionts Affect Vector Health and Reproduction and Contribute to Fitness of the Vector Host, Figure 6A,B). These bacteria exhibit a well-developed redox system, an important mechanism to survive the blood feeding process due to the high oxidative stress microenvironment during blood metabolism [28]. Similarly, in the mosquito, *A. gambiae*, Enterobacteriaceae and Flavobacteriaceae predominate in the midgut after blood feeding [24]. A similar phenomenon occurs in many insects, although the types of bacteria that survive are different than those found in ticks. In the kissing bugs, e.g., *Triatoma brasiliense*, certain enteric bacterial genera predominate in the midgut, primarily *Gordonia, Serratia, Mycobacterium* and *Rhodococcus*. In these bacteria, oxireductases comprise the majority of their enzyme components and also have genes coding for heme oxygenase, which catalyzes the degradation of heme and liberated iron, an essential nutrient for most bacteria [75]. These midgut dwelling bacteria also have a large number of enzymes involved in oxygen and nitrogen processing capabilities, providing an ability to degrade complex aromatic ring compounds, as well as acetate CoA ligase to TCA (tricarboxylic acid cycle) and energy metabolism. Other oxidoreductases, especially nitrate reductase and oxygenases, may also contribute to the ability of these bacteria to survive in the freshly blood-filled midgut, which may last for lengthy periods since ticks typically digest their blood meals rather slowly [76].

In the Lyme disease tick, *I. scapularis*, numerous environmental bacteria are acquired from soil and vegetation where these ticks survive off the host, or from the host skin during attachment for blood feeding. Many of these environmental bacterial genera are dependent upon the tick host for their blood meal nutrients but are lost during the transition from one juvenile stage to the next or from the juvenile stages to the adult female [53]. Less diversity was found among intracellular endosymbionts of this tick species, as shown in a study in one region of New York state where the microbiota was found to be dominated by a single genus and species, namely, *Rickettsia buchneri*. In adult females, the genus *Rickettsia* constituted 97.9% of the microbiota in this life stage [21]. In contrast, adult male ticks had a much lower abundance of this *Rickettsia*, ranging from 55–83%, but also supported a greater diversity of other bacterial genera than females, perhaps because males of this tick species do not blood feed. Field collected ticks exhibit a substantially greater diversity of bacterial endosymbionts than laboratory-reared colonies. However, the composition of the microbial community is also dependent upon geographic locations, as determined by sampling of wild-caught adult *I. scapularis*, and, especially important, varies greatly among individual tick specimens, even among those collected in the same region. In some regions of North Carolina, species of the Enterobacteriaceae predominated, whereas in many other specimens collected in this state *Borrelia* and *Sphingomonas* were predominant. Rickettsial endosymbionts predominated in most samples from Virginia, South Carolina and Connecticut/New York, but numerous individual specimens exhibited greater bacterial diversity of enteric symbionts (e.g., *Borrelia*, *Sphingomonas* and several genera of Enterobacteriaceae) with few or no rickettsial endosymbionts [77]. Another complicating factor appears to be the diverse hosts on which *I. scapularis* feed, ranging from lizards, ground feeding birds and numerous mammal species. Comparison of the abundance of the different species of microbial endosymbionts showed that in blood-fed *I. scapularis* nymphs, the relative abundance of *R. buchneri* varied from >65% of the microbial population in nymphs that fed on opossums (*Didelphys virginiana*) to only ~ ±5% in nymphs that fed on raccoons or grey squirrels. Similar extreme variations were observed for other common endosymbionts [22]. A similar phenomenon occurs in *Ixodes persulcatus* (Asian vector of Lyme disease bacteria). In unfed adults, the most abundant bacterial genera were *Acinetobacter* (21.58%), *Rickettsia* (18.95%), *Pseudomonas* (6.35%, *Chryseobacterium* (2.82%), *Sphingobacterium* (1.20%) and *Brevundimonas* (1.12%). Except for *Rickettsia*, the other surviving bacteria are enteric symbionts living in the midgut. Numerous other genera were identified but their abundance was <1% and these brief transients can be ignored. Following feeding, these proportions changed greatly. The most abundant genus was *Proteus* (33.44%), followed by *Rickettsia* (4.92%), *Morganella* (2.29%), *Comamonas* (1.94%), *Acinetobacter* (1.88%) and *Halomonas* (1.86%). All but *Rickettsia* are enteric bacteria, a finding consistent with the evidence described previously (Section 4), suggesting that their contributions to vector health is a factor in the survival of these ticks. The greatest decline was in *Acinetobacter*, from the most abundant genus in unfed ticks to one of the least abundant in fed ticks [20]. Clearly, blood feeding has a profound effect on the composition of the midgut microbiota, depending upon tolerance of ROS-related conditions and changes in nutrient composition [6,53,74,78].

In the Rocky Mountain wood tick, *Dermacentor andersoni* (vector of *R. rickettsii*), the core microbiome of the midgut in adult males comprises *Arsenophonus* (62.1%), *Francisella* (30.0%) and *Rickettsia* (7.7%); others present (≥1%) were *Ralstonia* sp. and species of Oxalobacteriaceae and Burkholderaceae. However, following blood feeding and reproduction, these proportions changed over three successive generations. By the third generation, *Francisella* increased to 65.3%, while *Arsenophonus* declined to 8.8% and *Rickettsia* to only 2.6%. *Arsenophonus* predominated in the salivary glands (98.2%) in the first generation but declined to 39.1% by the third generation and a new bacterium, *Acinetobacter*, became predominant (44.3%). The authors suggest that these generational changes may reflect the host’s vulnerability to contamination by environmentally acquired bacteria, especially *Acintetobacter*, commonly found in soil. Antibiotic treatment (Oxytetracycline) had little effect on midgut microbiota but greatly altered the microbiota of the salivary glands, where *Acinetobacter* reached 97.7% of the microbial community. Antibiotic treatment affected tick molting, survival and significantly reduced reproductive fitness, suggesting an important role for the microbiota in selected organs [79].

The greatest changes in microbial diversity occur during complete metamorphosis from larvae to adults. In holometabolous insects that have distinct larval, pupal and adult life stages, the digestive tract undergoes extensive remodeling that removes the gut contents, un-digested residues and bacteria, all incorporated in the peritrophic matrix and expelled as a meconium from the pupa, so that few or no bacteria remain in the emerging adults. Exceptions involve heritable bacteria that survive in specialized intracellular bacteriocytes [80]. In the sand fly, *Lutzomyia longipalpis*, the greatest bacterial diversity occurs in the larvae. Sand fly larvae, which feed on fecal wastes and plant materials, acquire microbes from the soil, although others may have been passed intergenerationally. In unfed adults, the intestinal microbiota was reported to comprise 57 genera, the most prevalent of which included *Acinetobacter, Stenotrophomonas, Pseudomonas, Flavimonas, Enterobacter, Klebsiella, Bacillus, Staphylococcus, Serratia, Yokenella, Burkholderia, Citrobacter, Escherichia, Pantoea, Morganella* and *Weeksella*. Following blood feeding, diversity diminished greatly [81] but eventually recovered in uninfected flies [10]. The microbiota of the blood-fed insects consisted mostly of a core group of at least 10 different symbionts that increased in abundance during or after feeding, specifically, representing the following six families: Moraxellaceae (*Acinetobacter* and *Enhydrobacter*), Enterobacteriaceae (*Enterobacter, Serratia, Pantoea*), Xanthomonadaceae (*Stentorophomonas*), Pseudomonadaceae (*Pseudomonas*), Bacillaceae (*Bacillus*) and Staphylococcaceae (*Staphylococcus*), as well as *Chryseobacterium*. In ticks, several endosymbionts remain as the predominant microbes in adults after blood feeding. In the Lyme disease tick, *I. scapularis*, two endosymbiont genera, *Rickettsia* and *Anaplasma*, and one enteric genus, *Borrelia*, were found to predominate [21]. In a survey of seven North American ticks, the authors found *Coxiella, Francisella, Rickettsia, Midichlori* and *Arsenophonus* as the most abundant endosymbionts [82].

## 6. Does the Vector Microbiome Affect the Survival and/or Development of Pathogenic Microbe Hosts?

In addition to their influence on vector health as described in the preceding sections of this review, symbiotic microbes also affect the survival of microbial pathogens in their insect and/or tick hosts and their transmission to susceptible vertebrates. Examples from the different insect and tick vector taxa follow below.

Two genera, *Phlebotomus* and *Lutzyomia*, are the primary vectors of *Leishmania* parasites. Species of *Phlebotomus* occur only in the Old World, whereas species of *Lutzomyia* occur only in the New World. The parasite’s developmental cycle takes place entirely in the midgut of the sand fly host and it is subject to the influences of resident microbiota. In these insects, e.g., *Phlebotomus duboscqi*, the gut microbiota are essential for the survival of the leishmaniasis pathogen, *Leishmania major*. Midgut microbes were reported to promote optimal osmotic conditions for the survival of the infective stages of the promastigotes. When the flies were subjected to blood meals laced with antibiotics, the infective metacyclic promastigotes could not colonize the anterior midgut and form the bolus-like mass normally injected into the human or animal host during blood feeding [83]. In the sand fly *L. longipalpis* infected with *Leishmania infantum*, bacterial diversity diminished while species of Acetobacteriacae became the dominant microorganisms in the vector’s digestive tract, coordinately with the increases in parasite abundance. When treated with an antibiotic cocktail (penicillin, gentamicin and clindamycin), *Leishmania* parasites were unable to undergo metacyclogenesis and complete their development to the infective stages [10]. These findings clearly show that Acetobacteriaceae are essential for *Leishmania* development and transmission. Other bacteria are harmful to the parasites, e.g., certain strains of *Serratia marcescens* that produce a compound called prodigiosin. This compound has recently been shown to be toxic to *Leishmania mexicana* by disrupting their mitochondria and triggering programmed cell death of the parasites [17]. The extent to which such interplay between the bacteria that exert toxic effects that might interfere with the dynamics of *L. infantum* or other *Leishmania* sp. transmission awaits further investigation. In addition to their effect on vector health, midgut bacteria (*Tsukamurella, Lysinibacillus, Paenibacillus, Solibacillus* and *Bacillus*) egested by the sand flies during blood feeding into the host skin was reported to prime the resulting inflammasome, leading to high levels of IL1B that recruit neutrophils to the bite site. Remarkably, neutrophils appear to shield the invading *Leishmania* parasites and contribute to their successful invasion of the vertebrate host [84].

In *Anopheles* mosquitoes examined in several locations in Zambia, the presence of *Enterobacter* sp. inhibited the ability of *Plasmodium falciparum* to invade the insect’s midgut epithelium as a result of genes coding for ROS-related enzymes and compromised development of the parasites in the mosquito host [9]. Haemolysins (cytolysins) and the bacterial toxin prodigiosin produced by *Serratis marcescens*, *Enterobacter* sp. and other enteric bacteria are lethal to malaria parasites, e.g., *P. falciparum* [85], and also to *Glossina* [86].

In ticks, many species of *Coxiella, Francisella* and *Rickettsia* bacteria are mutualistic symbionts [14], although several representative species are pathogenic in the tick’s mammalian hosts. In these vectors, infection with one species of the Rickettsiales has been shown to block infection with a similar but different rickettsia species. The presence of *R. peacockii*, an endosymbiont prevalent in the Rocky Mountain wood ticks, *Dermacentor andersoni*, prevents infection with the pathogen *R. rickettsii* [87,88]. The same phenomenon was shown to occur in the American dog tick, *D. variabilis*, where infection with the common endosymbiont, *R. montanensis*, prevents infection with a different rickettsia, *R. rhipicephali* [89]. Another example occurs in the lone star tick, *A. americanum*, where the presence of *Coxiella*-like endosymbionts in the salivary glands impairs transmission of *Ehrlichia chaffiensis*. As noted previously, these endosymbionts are critical for the fitness of lone star ticks [58]. The endosymbiotic bacterium *Francisella* is an important microbe for the survival of its tick host, *D. andersoni*. However, certain *Francisella* endosymbionts infecting these ticks were reported to suppress the tick’s innate immune system, thereby providing a more favorable environment for multiplication of the pathogen *F. novicida*, although its abundance differed in different geographic regions. In addition, pathogen prevalence was found to be tissue specific, differing between the midgut and the salivary gland, over multiple generations. Consequently, the effects of competition between these different *Francisella* species may occur through the tick’s geographic range [90].

Chigger mites present another example of pathogen suppression of other competing microbes. This was found in *Leptotrombidium imphalum*, where infection with the scrub typhus rickettsia, *O. tsutsugamushi*, together with a species of *Amoebophilaceae* (bacteria) profoundly reduced the abundance of all other microbiota [62].

In ticks, transovarial transmission of tick-borne pathogens and non-pathogenic endosymbionts is a common phenomenon [87,91], although the factors that regulate this process are largely unknown. A surprising finding was the discovery of elevated densities of *R. montanensis* in the ovaries of female *D. variabilis* following treatment with 20-hydroxyecdsysone (20-E) [33], increasing 6.8-fold in density in the ovaries while decreasing 3.5-fold in the carcasses. Injection of 20-E into tick females during blood feeding was shown to trigger vitellogenesis and vitellogenin uptake by receptors on the developing oocytes [92]. The authors speculate that rickettsiae may invade the ovaries by “hitchhiking” onto the vitellogenin/vitellogenin receptor transport system. However, increases in the densities of *Francisella* sp., also present in the tick ovaries, in response to this same stimulus suggest that B vitamins and other nutrients provided by these bacteria may explain how they benefit the tick host.

## 7. Conclusions

In this review, we examined the microbiomes of a wide range of blood-feeding vectors, including diverse insects and acarines. At first glance, there appears to be immense diversity in the hundreds of bacterial genera present in the different vector species. However, we showed that despite their very different evolutionary origins, these blood-sucking disease vectors, including sand flies, mosquitoes, kissing bugs, tsetse flies, fleas, lice and ticks all share a common core constituency of at least eight bacterial families. These families include the Bacillaceae, Rickettsiaceae, Anaplasmataceae, Sphingomonadaceae, Enterobacteriaceae, Pseudomonadaceae, Moraxellaceae and Staphylococcaceae. The microbiomes of several of these vectors exhibit an extraordinary diversity and abundance of bacterial species (so-called species richness), e.g., the sand fly *L. longipalpis*, with 110 different genera comprising 86 different families, 19 different classes and nine different phyla, while others exhibit an unusual paucity of microbial symbionts (e.g., tsetse flies, *Glossina fuscipes*, with only 22 genera comprising 14 families). There is considerable overlap in shared bacterial genera between different insect groups, e.g., between the mosquito *C. quinquefasciatus* and the sand fly *L. longipalpis*, but much less between either of these insects and a representative tick, *I. scapularis*.

The reasons for the persistent occurrence of these bacterial symbionts are not fully understood but likely relate to their contribution of essential nutrients found lacking in vertebrate blood, the food source of the blood-feeding vectors. Consequently, we examined 21 genomes of the most common bacterial symbionts (www.patric.org, accessed on 9 May 2021), comparing blood-feeding vectors versus non-blooding insects. We found that certain enteric bacteria benefit their hosts by upregulating large numbers of genes coding for essential nutrients, especially B vitamins, including biotin, cobalamin, flavin, folate, pimelate, pyridoxal, pyridoxine, pyrimidine, riboflavin, thiamine and xanthine/uracil. Blood-sucking vectors expressed highly significantly more genes (*p* < 0.001) coding for these essential nutrients than non-blood-feeding insects. Moreover, the genomes of 12 different enteric bacteria contained significantly more genes (*p* < 0.001) coding for these vitamins than the genomes of six different endosymbiotic bacteria, as well as for genes coding for reactive oxygen species or essential amino acids. In contrast, non-blood-feeding insects expressed few or none of the same gene families.

Bacterial diversity was often greatest in the juvenile stages of the blood-feeding vectors but declined greatly as the hosts matured, and even more so after they had ingested their blood meals. However, some bacteria, e.g., Acetobacteriaceae in sand flies, remained and proved essential to host growth and transmission of their *Leishmania* sp. parasites. Other bacteria, e.g., *Serratia marcescens*, were lethal for *Leishmania* but not for the sand fly hosts. Similar findings of essential symbiotic bacteria have been reported for mosquitoes, where several enteric bacteria are lethal to malaria parasites, or ticks, where the presence of *Coxiella*-like endosymbionts in the salivary glands impairs transmission of *Ehrlichia chaffiensis*. Greater understanding of the interrelationships between the numerous symbiotic bacteria in the complex microbial communities and the different pathogens transmitted by these blood-feeding vectors may lead to novel methods for preventing disease transmission.

## Figures and Tables

**Figure 1 microorganisms-09-02433-f001:**
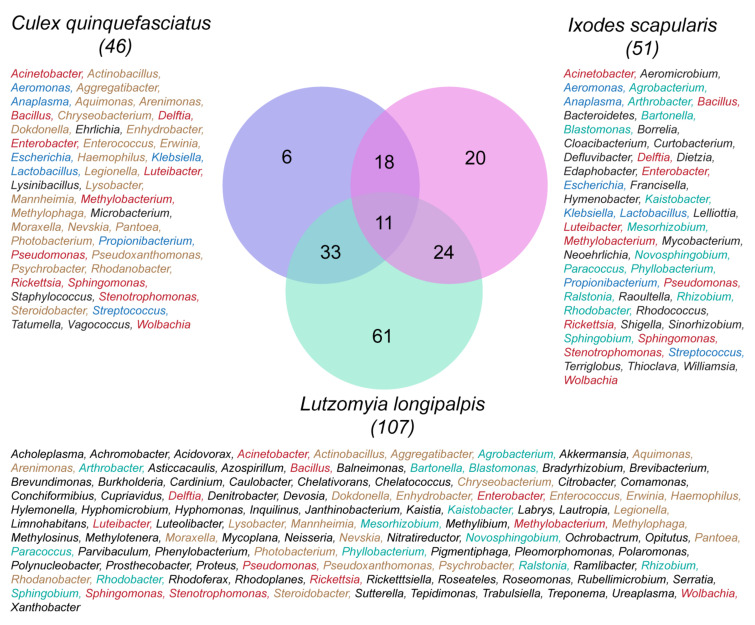
Venn diagram (iVENN program, created at RML) comparing the variety of genera of symbiotic bacteria and the extent to which they are shared among two insect and one tick species—representative species of vector insects and ticks: *Culex quinquefasciatus,* with 46 genera; *Lutzyomia longipalpis,* with 107 genera, and *Ixodes scapularis,* with 51 genera. *C. quinquefasciatus* has six unshared genera (shown in black font), but also shares 33 of its genera with *L. longipalpis* (brown font), 11 with both *L. longipalpis* and *I. scapularis* (red font) and 18 exclusively with *I. scapularis* (blue font). *L. longipalpis* has 61 unshared genera and 24 others shared exclusively with *I. scapularis* (turquoise font); *I. scapularis,* with 51 genera, shares 11 with both insects, as noted previously. Colors indicate level of sharing: red (shared by all three vectors); blue (shared between mosquito and tick); brown font (shared between mosquito and sand fly); turquoise (shared between tick and sand fly).

**Figure 2 microorganisms-09-02433-f002:**
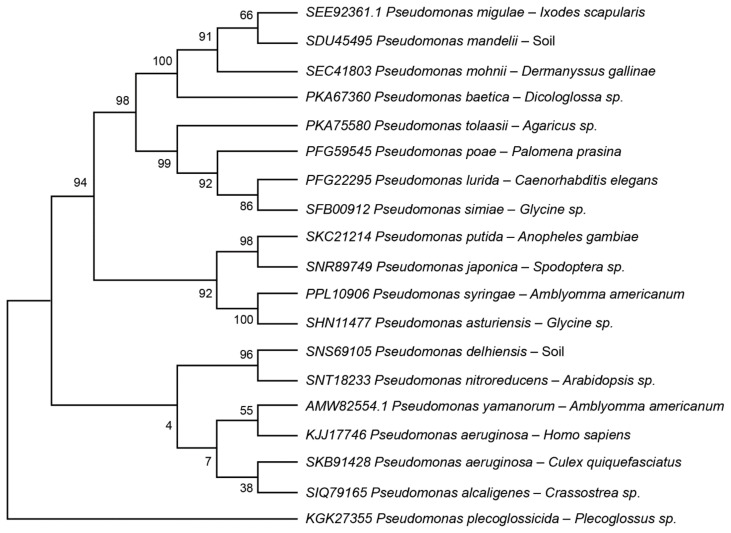
Comparison of the bacterial protein peptidoglycan synthetase in 19 different members of the genus *Pseudomonas*. Phylogenetic tree constructed using the maximum likelihood method. Accession numbers and source of isolation (i.e., vectors, hosts or environment) are indicated.

**Figure 3 microorganisms-09-02433-f003:**
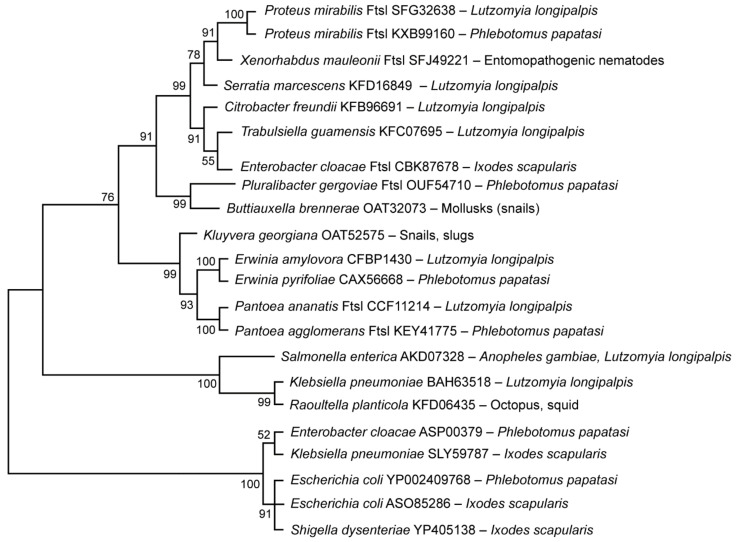
Comparison of the bacterial protein peptidoglycan synthetase in 22 members of the family Enterobacteriaceae (amino acid alignment in Appendix A). Phylogenetic tree constructed using the maximum likelihood method. Genbank accession numbers and respective vectors/hosts follow the scientific names of the bacteria.

**Figure 4 microorganisms-09-02433-f004:**
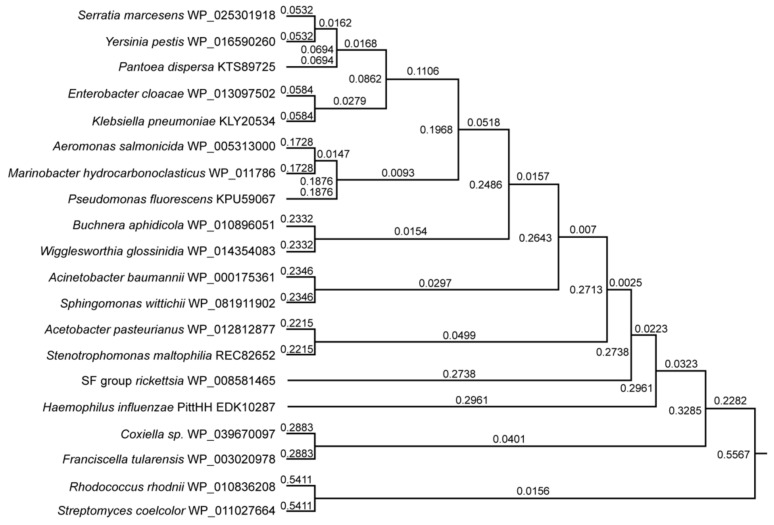
Phylogenetic analysis of biotin synthetase among 20 different bacterial species. Alignment (Clustal Omega, not shown) of these sequences showed an overall similarity of 71.6% (Blossum 62). Except for certain outliers, e.g., *Streptomyces coelcolor* and *Rhodococcus rhodnii* (which grouped closely with each other), most of the others aligned more closely, indicating that despite the wide variety of bacterial families, biotin synthetase is a highly conserved molecule in these bacteria.

**Figure 5 microorganisms-09-02433-f005:**
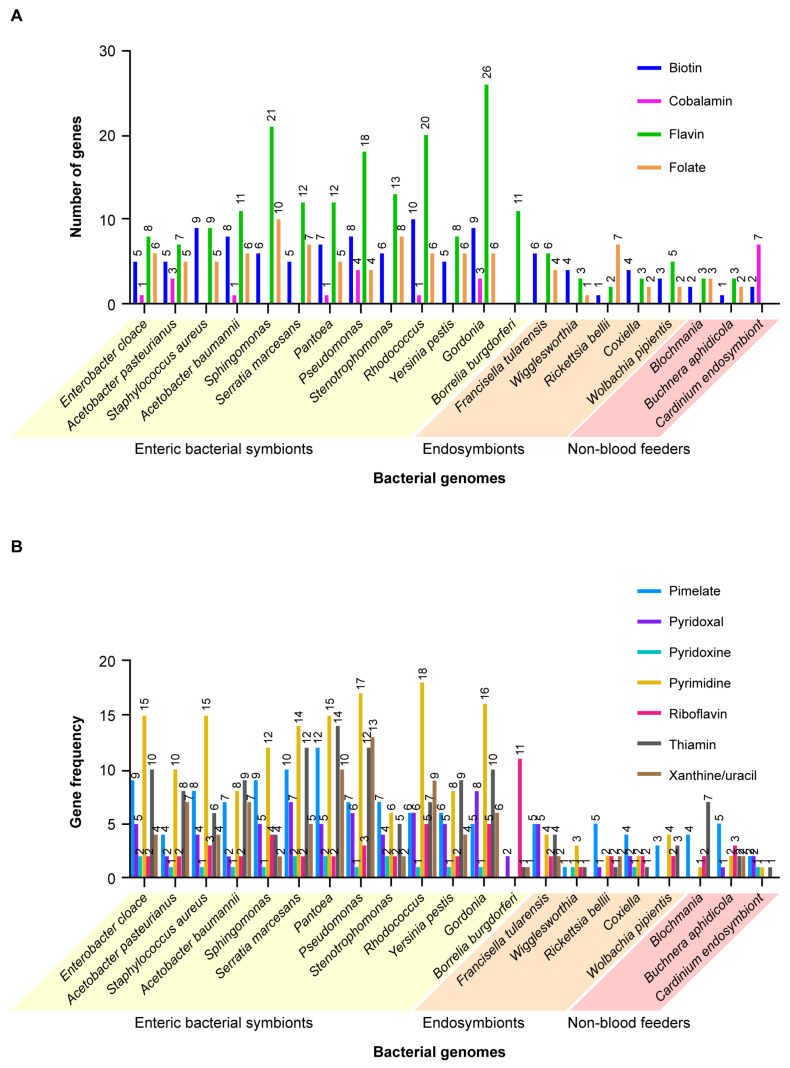
Comparison of the number of genes coding for different vitamins expressed by 18 symbiotic bacteria in blood-feeding insects and ticks versus three symbiotic bacteria in plant-feeding insects. Differences between enteric bacterial symbionts versus endosymbionts also are compared. (**A**) Biotin, cobalamin, flavin and folate. (**B**) Pimelate, pyridoxal, pyridoxine, riboflavin, thiamin and xanthine/uracil. Results show bacterial symbionts (both enteric and endosymbiont) in blood-feeding arthropods express significantly more genes coding for diverse vitamins than non-blood feeding insects.

**Figure 6 microorganisms-09-02433-f006:**
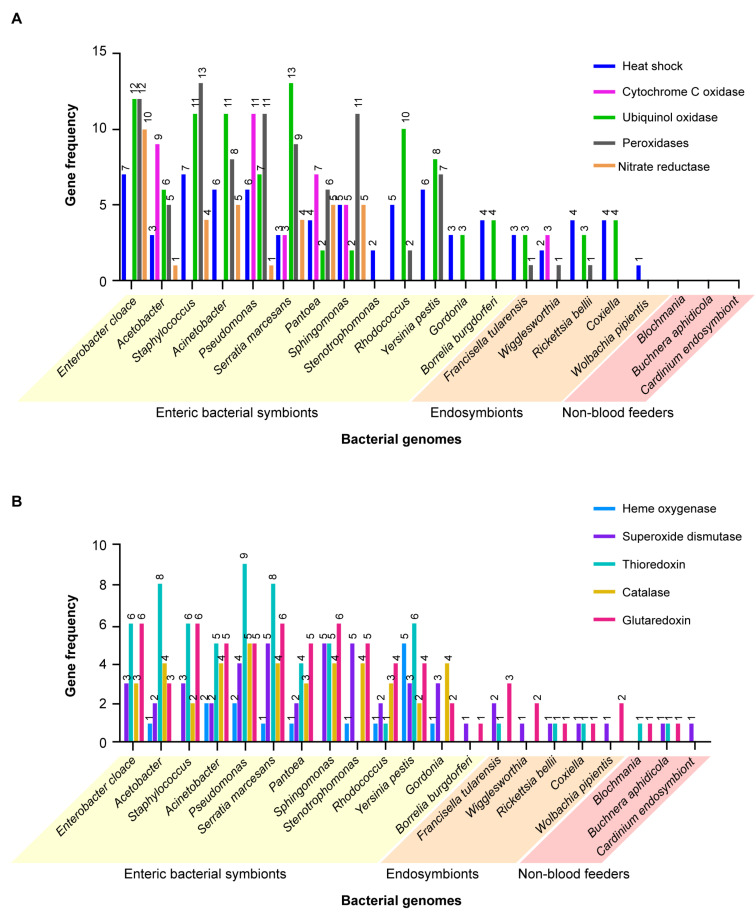
Comparison of the number of genes coding for different proteins for coping with various cell stressors expressed by 18 symbiotic bacteria in blood-feeding insects and ticks versus three symbiotic bacteria in plant-feeding insects. (**A**) Heat shock protein (mostly HSP60, HSP70 and HSP90), cytochrome C, ubiquinol, peroxidase and nitrate. (**B**) Heme oxygenase, superoxide dismutase, thioredoxin, catalase and glutaredoxin.

**Figure 7 microorganisms-09-02433-f007:**
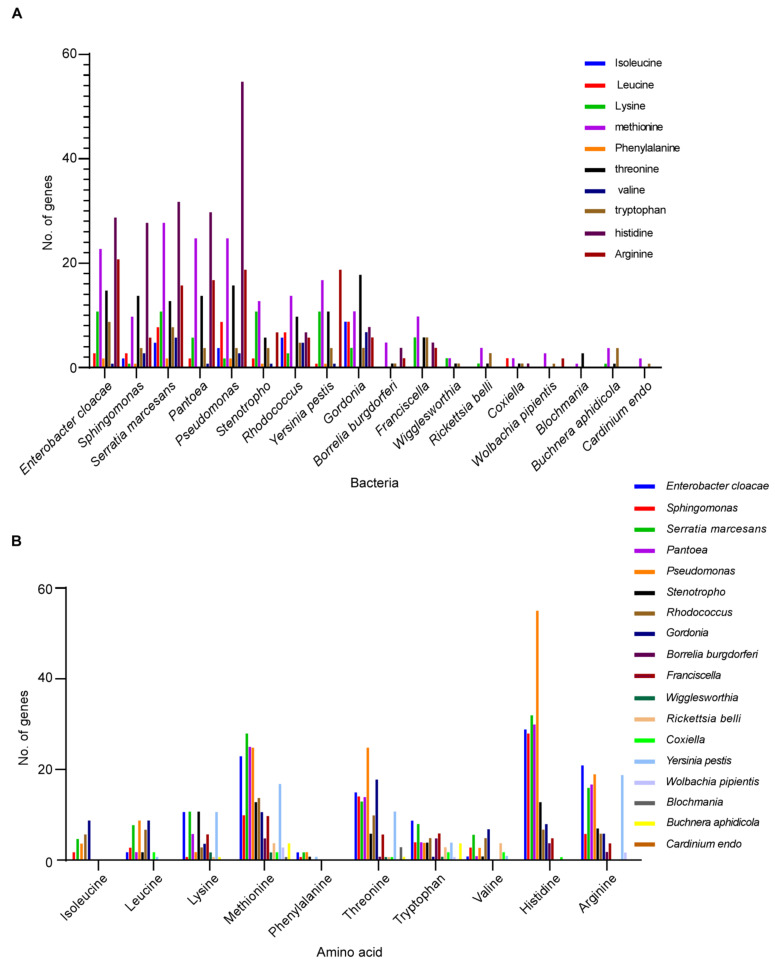
Graphs showing comparison of abundance of genes coding for ten essential amino acids expressed by symbiotic bacteria in blood-feeding insects and ticks versus symbiotic bacteria in plant-feeding insects. (**A**) Bacteria species along the axis versus abundance of genes for the different amino acids on the ordinate. (**B**) Amino acids along the axis versus number of genes forming the ordinate.

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
