# Peer review of "Microbiomes of Blood-Feeding Arthropods: Genes Coding for Essential Nutrients and Relation to Vector Fitness and Pathogenic Infections. A Review"

_microorganisms, 2021, doi:10.3390/microorganisms9122433_

Round 1

Reviewer 1 Report

The manuscript by Sonenshine and Stewart entitled “Microbiomes of blood-feeding arthropods: genes coding for essential nutrients and relation to vector fitness and pathogenic infections. A review” analyzes the information available through the literature and public databases about microbiota meta-omics of blood-feeding arthropods, in order to highlight the possible existence of a microbial core and its relationship with host fitness and its ability to vector vertebrates’ pathogens.

I find the topic very interesting, and reading the abstract I was attracted by it (even though I found it could be improved, as below indicated), but the text is hard to follow for several reasons below explained. Therefore, I consider that the manuscript does not fulfill the expectative and cannot be published in its current format. It should be thoroughly reviewed and needs to be rewritten in many parts to correct formal and experimental defects before being acceptable for publication.

First of all, it surprises me that the manuscript is presented as a review (both in the article type and in the title), taking into consideration that it includes “Materials and Methods” and “Results and Discussion” sections. Obviously, even though the data used for the analyses have not been generated by the authors and come from bibliographic searches and sequence databases, the informatics and statistical methods employed generate novel results to test the authors hypotheses and, therefore, it should be considered a Research article.

Second, it seems that the manuscript has been submitted in a rush, as there are many formal defects that can be easily solved with an accurate revision before submission. A few examples: taxonomic nomenclature is not always written in italics; English usage is not always optimal, even though both authors should be fluent in English; Supplementary material seems to be uncarefully prepared.

Third, the aim of the work is difficult to identify at the beginning. The abstracts is written in a confusing manner. I do not find appropriated to write a direct question in the second sentence, which is not the question that is answered in the next sentence, nor is it the only question posed in the manuscript. It is not clear either if they focus on arthropods feeding on mammalian only or terrestrial vertebrates, not even if all kind of blood-feeding arthropods are included in the study or only a few groups inside the two wide categories of insects and acarines (e.g., see Balashov, 2006, 10.1134/S0013873806080112, for a summary of the different groups in which this kind of feeding behavior has been described).

However, the main flaws of the manuscript refer to the way symbiosis is defined, the selected examples for comparison, and the way the results are presented for the analyses.

Symbiosis definition. Throughout the text, it is not clear if endosymbionts are included in the definition of “microbiota”. Considering the high differences in the presence of ecto-endosymbionts in different insect species (the field I am familiar with; see, as an example, Shan et al. 2021, https://doi.org/10.3390/microorganisms9020464), it would be necessary to clarify it from the beginning. Furthermore, based on the different ecology and genome evolution, it does not seem appropriate to compare directly endosymbionts and enteric ectosymbionts. It would be necessary to take also into account the genome size (obviously free-living or cultivable microorganisms have larger genomes, therefore more metabolic genes, whether expressed/useful-to-the- host or not, - just because they need them for survival outside the host.  In addition, it is not clear how endosymbionts are defined (Yersinia pestis is an endosymbiont? It appears as such in Figures 5 and 6).

Comparisons.  First, when using an “outgroup” they consider a honeybee parasite… Why only an acarine? Why an invertebrate’s pathogen instead of a vertebrate’s one? Why not a plant-parasite? It seems to me that there are too many uncontrolled variables to be able to extract clear conclusions.

The authors present lots of phylogenetic trees for proteins that seem to be highly conserved in all groups analyzed, since there is no clustering by taxonomic lineages. But it is not clear if the bacteria analyzed in this context are specific of the host or are the same that can be found in the environment. Furthermore, it is not possible to evaluate if the same similarities are found in free-living bacteria of the same taxonomic groups. Therefore, no conclusions can be drawn from these results.

The way the results are presented.

Figure 1. The colors on the legend are unclear, as many genera are shown in black or red font in all three circles, and there are genera in blue font both in Culex and Iodes.

Figure 2. Panel A cannot be read and it is unnecessary, unless you present only relevant regions. The text indicates that the 19 species of Pseudomonas have been found in different vector insects and ticks, but panel B of the figure indicates that the vector is unknown in most of the cases. Please, explain. Could it be possible that they are just acquired horizontally from the environment and, therefore, have no relationship with the host and, therefore, no conclusion can be drawn from this results?

Figure 3. The two panels show phylogenetic trees; no sequence alignment is shown (which, otherwise, will be unnecessary, as in the previous figure). Species names should be presented complete or abbreviated, but in a consistent manner.

Figure 4. Why the tree is presented in the opposite direction? Different classes or families should be presented in different colors, to show that there is no clustering by family.

Figure 5. I don’t understand the way the different bacterial groups are presented. Is Yersinia pestis an endosymbiont? And it seems unclear why it is not indicated that the bacteria presented from non-blood feeders are also endosymbionts, nor if they are non-blood feeders that have enteric bacteria, or if the enteric and endosymbiotic bacteria are present in the same host or in different ones. Furthermore, it is not informative to present just the number of genes (total number of genes detected through metagenomics or expressed genes detected through metatranscriptomics?) when the compared bacteria are so different. Long-term endosymbionts have very small genomes in which many metabolic genes have been lost because the products can be obtained from the host, but the few preserved genes tend to be crucial for the maintenance of the whole symbiotic system. Taking into account that the number of genes in each compared category is quite small, it would be more interesting to prepare a table, to see which are the shared or specific genes, and also some figure indicating if the relevant/essential metabolic pathways are complete or not, and if different ectosymbionts participate together. It would be also interesting to see if the missing steps can be performed by the host, which has been shown to perform the final steps in many endosymbiotic systems, as a way of ”controlling” the endosymbiont. Additionally, it is necessary to indicate which bacterial strain is used on the comparison, as well as the accession numbers of the genomes used in the compassion, because many different strains for each bacterial species have been studied, and in some cases the genome size differences are quite big, while each strain can have different genes acquired through horizontal gene transfer.  

Supplementary material is not well structured.

Supplemental Figure S1 does not provide any relevant information, it seems to be just a snapshot of the COG web page (which, furthermore, is not referenced). All other Supplemental figures (except for S4, which presents an alignment) are not figures but tables. Supplemental Figure S2 seems that has not been curated, as one column is called “Remove?”.

Supplementary Table S1: rows should be ordered in a way easy to identify, e.g., alphabetically.  The way the references are presented is confusing and seems disorganized. It would be interesting to group the columns (and indicate so) based on arthropod lineages. It would be interesting to include the microbiota of other non-blood feeding insects, such as cockroaches, lepidoptera…

Supplementary Table S2: What does SigP/SP/sp mean? First row contains enzymes, not metabolites.

Below are some additional comments:

Introduction

Deep pyrosequencig is no longer in use since Roche discontinued the 454 sequencing platform in 2013. It would be better to refer just to the NGS technology to include all second-generation massive sequencing.

The proper name of the domain analyzed in this work is “Bacteria”. “Eubacteria” was a name given in the past to one of the kingdoms inside the domain Bacteria, but (to my knowledge) it is not a currently valid taxonomic range.

The final part of the introduction is confusing. The same questions (more or less) posed at the second part of page 3 in a row are, immediately after, posed with numbered entries. It would be better to present them in a single paragraph, and clearly define the titles of the subsections to be presented as results.

Materials and methods. Databases and Methods employed should be described in more detail, indicating the programs used for each purpose.

Results and Discussion

When talking about tissue tropisms, the authors mention several vertebrate pathogens but identify them as “mutualistic” for the arthropod. Which evidence have they of such kind of symbiosis?

There are too many places in which the authors indicate that the details of what they are presented in one section are better presented and discussed elsewhere in the manuscript. This difficult a fluent reading.

I read in the manuscript that Wolbachia is used as an example of bacteria that is present only in male testes. I could not find the reference to this finding and, to my knowledge, Wolbachia infects the ovaries and is maternally transmitted. Please, clarify.

Section 3.2.1 is confusing. It appears just a disorganized sum of examples. Is the last paragraph of the section the summary of all section?

Genes do not code for vitamins or nutrients, but for proteins. The correct way to mention it would be genes coding for proteins involved in biosynthetic pathways.

A list of identified bacteria does not give information about the microbiota complexity if their relative abundance is not also indicated. And both parameters need to be used in the comparison among hosts.

Finally, I recommend the authors to include numbered lines in their manuscripts to facilitate the review process.

Reviewer 2 Report

Italicize scientific names throughout page 2: "The microbiomes comprise protozoans and fungi as well as Eubacteria, Archea, viruses and eukaryotic microbes." Saying eukaryotic microbes seems somewhat redundant after having listed protozoans and fungi. Maybe add "other" in front of "eukaryotic" or just delete "eukaryotic microbes". And change "Archea" to "Archaea". page 3: "Evidence for compared the shared bacterial taxa was done using the Venn diagram tool...." delete "compared" page 3: “The honeybee parasite (Varroa destructor) is included as a non-blooding feeding contrast.” Change to “non-blood” a non-blood feeding arthropod contrast: would be interesting to look at lice, Anoplura vs. those Mallophaga that do not consume blood Figure 7A, I think the y-axis is meant to be labeled as “No. genes”? Thank you for an important and interesting review!

Author Response

We thank this reviewer for his/her very supportive comments.

  1. We have italicized all the scientific names. This was something that occurred during the journal’s uploading process.  Italics were somehow omitted from the first half of the manuscript. We checked and found that all scientific names were italicized in the original submission. We have gone through the manuscript and restored all the italics there omitted and hope that the submission process of this revised version does not remove the italics again.
  2. Re: repeating eukaryotic microbes is redundant.  Yes, we agree and we have deleted it (line 85). We retain the next sentence saying that this manuscript is focused on bacteria. Thank you for the correction.
  3. RE: misspelling of Archaea, we corrected it (line 85). Thanks for improving our spelling.
  4. On P. 3 we deleted “compared” and substituted “comparing”. We have restructured this sentence and we appreciate the grammatical correction.
  5. On P. 3, we changed “non-blooding” feeding to “non-blood feeding” (line 120). Again, thanks.
  6. Re: adding another outlier, we introduced the mite, Dermanyssus gallinae, a different acarine (although also a blood feeder) in Table S1. It is a well-studied parasite and with considerable study of its microbiome.  We considered also adding biting lice (Mallophaga) but there was little substantial information about its microbiota. Thank you for the suggestion.
  7. RE: Figure 7A, yes, the Y-axis should be “No. genes”.  We have amended the figure accordingly. Thank you.

Round 2

Reviewer 1 Report

To avoid changes during the process of submitting the comments, you can find them in a pdf file.
